# Shorter Axial Length Is a Risk Factor for Proliferative Vitreoretinopathy Grade C in Eyes Unmodified by Surgical Invasion

**DOI:** 10.3390/jcm10173944

**Published:** 2021-08-31

**Authors:** Sakiko Minami, Atsuro Uchida, Norihiro Nagai, Hajime Shinoda, Toshihide Kurihara, Norimitsu Ban, Hiroto Terasaki, Hitoshi Takagi, Kazuo Tsubota, Taiji Sakamoto, Yoko Ozawa

**Affiliations:** 1Department of Ophthalmology, Keio University School of Medicine, 35 Shinanomachi, Shinjuku-ku, Tokyo 160-8582, Japan; saki.love5@icloud.com (S.M.); uchidats@gmail.com (A.U.); nagai@a5.keio.jp (N.N.); shinoha@mac.com (H.S.); kurihara@z8.keio.jp (T.K.); nban@keio.jp (N.B.); tsubota@z3.keio.jp (K.T.); 2Department of Ophthalmology, St. Marianna University School of Medicine, Kanagawa 216-8511, Japan; htakagi@marianna-u.ac.jp; 3Laboratory of Retinal Cell Biology, Keio University School of Medicine, 35 Shinanomachi, Shinjuku-ku, Tokyo 160-8582, Japan; 4Laboratory of Retinal Cell Biology, St. Luke’s International University, 9-1 Akashi-cho, Chuo-ku, Tokyo 104-8560, Japan; 5Department of Ophthalmology, St. Luke’s International Hospital, 9-1 Akashi-cho, Chuo-ku, Tokyo 104-8560, Japan; 6Department of Ophthalmology, Kagoshima University Graduate School of Medical and Dental Sciences, Kagoshima 890-8520, Japan; teracchi16@yahoo.co.jp (H.T.); tsakamot@m3.kufm.kagoshima-u.ac.jp (T.S.)

**Keywords:** retinal detachment, proliferative vitreoretinopathy, axial length, nationwide study, propensity score matching

## Abstract

Purpose: To investigate the risk factors for the development of proliferative vitreoretinopathy grade C (PVR-C), independent of prior surgical invasion. Methods: Patients who underwent surgery for rhegmatogenous retinal detachment were prospectively registered with the Japan-Retinal Detachment Registry, organized by the Japanese Retina and Vitreous Society, between February 2016 and March 2017. Data obtained from general ophthalmic examinations performed before and at 1, 3, and 6 months after surgery were analyzed. Results: We included 2013 eyes of 2013 patients (men, 1326 (65.9%); mean age, 55.2 ± 15.2 years) from amongst 3446 registered patients. Preoperative PVR-C was observed in 3.6% of patients. Propensity score matching revealed that a shorter axial length (AL) was a risk factor for preoperative PVR-C (OR (Odds Ratio), 0.81; 95% CI (Confidence Interval), 0.69 to 0.96; *p* = 0.015), which was a risk factor for surgical failure (OR, 4.22; 95% CI, 1.12 to 15.93; *p* = 0.034); the association was particularly significant for eyes with an AL < 25.0 mm (*p* = 0.016), while it was insignificant for eyes with an AL ≥ 25.0 mm. Conclusions: A shorter AL was related to the development of PVR-C before surgical invasion. Our results will help elucidate the fundamental pathogenesis of PVR and caution clinicians to meticulously examine eyes with a shorter AL to detect retinal detachment before PVR development.

## 1. Introduction

Recent advancements in surgical technology have improved the surgical outcomes of retinal detachment. However, proliferative vitreoretinopathy (PVR) [1] which is characterized by the growth and contraction of the cellular membrane on both surfaces of the detached retina [2] remains the most common cause of surgical failure [3,4]. The pathogenesis of PVR involves chronic inflammation, and postoperative PVR formation may be strongly associated with surgical invasion [5]. However, PVR can also occur preoperatively and in eyes that do not necessarily have a clear history of inflammation, such as uveitis. The discovery of the risks of PVR development in the absence of prior surgical modification may help elucidate the fundamental pathogenesis of PVR and facilitate the diagnosis of retinal detachment in the early stage, before PVR development, by focusing more attention on eyes at risk in daily clinical practice.

PVR develops by the proliferation of migratory cells, mainly the retinal pigment epithelium (RPE) and glial cells that diffuse through retinal breaks, and the production of extracellular matrix which forms on the membranes to cause retinal folds [6]. PVR is classified by clinical signs and can progress from grade A to C. Grade A is defined by a vitreous haze and pigment clumps, grade B is by wrinkling of the inner retinal surface, retinal stiffness, vessel tortuosity, rolled and irregular edge of a retinal break, and decreased mobility of the vitreous, and grade C is by full-thickness retinal folds and/or subretinal strands posterior to and/or anterior to the equator [1,7]. PVR formation is exacerbated by inflammatory cytokines with or without microglia/macrophage recruitment, following the breakdown of the blood–retinal barrier by retinal breaks and/or vitreous traction [6], especially when the lesions remain untreated for a certain period. Meanwhile, previous studies have shown that specific single-nucleotide polymorphisms can be predictors of PVR [8,9]. Thus, PVR is a multifactorial disease resulting from environmental factors related to retinal detachment and factors native to the individual. 

Herein, we performed a large-scale analysis of clinical data obtained from the Japan-Retinal Detachment Registry (J-RD Registry), maintained by the Japanese Retina and Vitreous Society (JRVS) [10,11,12,13]. The J-RD Registry, which was established referring to previous population-based studies [14], collected data from 3446 participants with retinal detachment, who were registered at 26 qualified institutions. In this study, we analyzed the risk factors of patients who exhibited PVR at baseline, i.e., before surgery, and focused on the conditions of retinal detachment and native features of the eyes, i.e., axial length (AL), with the potential to affect the posterior vitreous condition, with respect to liquefaction and detachment [15]. We previously reported that a shorter AL was related to relatively poor postoperative visual outcomes in patients with an idiopathic epiretinal membrane (iERM). A shorter AL is also related to a high prevalence of diabetic retinopathy among patients with diabetes [16]. While retinal detachment is often found in eyes with myopia or a long AL [10,17], we hypothesized that retinal detachment in eyes with a shorter AL would frequently have a severe manifestation, i.e., PVR. Since the pathogenesis of PVR involves chronic inflammation, eyes exhibiting preoperative PVR may have had a relatively long history of retinal breaks and/or detachment, irrespective of the identification of the symptoms. Clinicians may not always perform complete fundus examinations when patients complain of only minute changes, such as subtle floaters; however, they should meticulously examine eyes at risk for PVR, to detect the lesions before the development of PVR.

## 2. Materials and Methods

### 2.1. Data Source and Participants

We performed this study using the data collected in the J-RD Registry system [10]. Briefly, patients who underwent surgery for rhegmatogenous retinal detachment at 26 institutions in Japan (see Acknowledgements) were registered. All surgeries were performed by 145 surgeons certified by the Japanese Ophthalmological Society. Registration was performed between February 2016 and March 2017, and data collection was performed through a website dedicated to the study (https://secure2.visitors.jp/retinal_detachment/ (accessed on 9 August 2020). The Ethics Committee of Kagoshima University approved the main study protocol (140093, 28–38), and all facets of this study were conducted in accordance with the tenets of the Declaration of Helsinki. The requirement for acquiring individual written informed consent from the patients was waived by all hospitals and institutes, except for the Kyushu University Hospital Ethics Committee, as this was an observational study based on information collected during the course of standard care, and the patient information was deidentified in the registry.

### 2.2. Data collection

The detailed procedure for data collection and baseline characteristics of the patients has been described elsewhere [10]. The members of the JRVS registry committee developed the questionnaires enquiring about the presurgical clinical characteristics, surgical interventions, and postoperative clinical course based on previous studies [4,14,18]. The AL of the eye was measured using an optical biometry device for macula-on eyes and an ultrasonographic instrument for macula-off eyes [10]. The clinical data acquired 1, 3, and 6 months postoperatively were also collected as previously described [10]. 

### 2.3. Surgery

The surgical methods were determined by the individual surgeons. All patients were treated with pars plana vitrectomy and/or scleral buckling [11].

### 2.4. Definition of Primary Success and Failure

The outcome was defined as a primary success when no additional surgeries were required during the 6-month postoperative period, except for surgery to remove silicone oil, with no other procedures. The definitions of failure levels from 1 to 3 were adopted from the European vitreo-retinal society retinal detachment study [14]. Briefly, level 1 signifies the requirement of additional surgeries for recurrent retinal detachment after the initial surgery and that the retina is attached at month 6; level 2 signifies the requirement of a sustained silicone oil tamponade at month 6, and level 3 signifies the persistence of retinal detachment with or without additional surgeries at month 6; the most severe failure was level 3 [11]. Silicone oil removal was planned within 6 months if no other additional procedures were required, and the surgery to remove silicone oil was not counted as an additional surgery [11].

### 2.5. Statistical Analyses

Data were presented as the mean ± SD. We used Mann–Whitney U and chi-square tests for between-group comparisons. We performed logistic regression analysis to prepare the propensity score-matched cohort according to the presence or absence of retinal detachment exhibiting PVR grade C (PVR-C), diagnosed using the updated Retina Society Classification published in 1991 [1]. The model included the following variables pertaining to the pretreatment characteristics: age, sex, number of retinal breaks, type of break (holes/horseshoe tears or macular hole/dialysis), size (≤30 or >30 degrees) and location (superior or inferior/posterior polar retina) of the greatest retinal break, and area of retinal detachment (1–4 quadrants). We confirmed the success of the matching procedure using an area under the receiver operating characteristic curve of 0.879. We considered *p*-values < 0.05 to indicate statistical significance.

## 3. Results

Of the 3446 registered patients, we excluded 1134 patients in the registry who had missing data required for the current study, 274 patients who had past histories of ocular surgery other than cataract, 15 patients who had aphakic eyes, and 10 patients in whom retinal breaks had not been identified. One eye was registered per patient; thus, we included the data of 2013 eyes of 2013 patients in the analyses. The patients’ mean age was 55.2 ± 15.2 (range, 9–94) years, and 1326 eyes belonged to men (65.9%) (Table 1). Table 1 describes the characteristics of the retinal breaks and retinal detachment. Sixty-seven eyes (3.3%) had choroidal detachment at baseline. The mean AL was 25.6 ± 1.9 (range, 16.4–33.0) mm; 1941 eyes (96.4%) showed no PVR (PVR-N) or PVR-B, and 72 eyes (3.6%) showed PVR-C at baseline. 

We divided the eyes into two groups based on the absence or presence of PVR-C (Table 1). Horseshoe tears were observed less frequently (PVR-N or B versus PVR-C; 78.5% versus 63.9%, *p* = 0.003), while a macular hole (2.4% versus 6.9%, *p* = 0.018) or dialysis were observed more frequently (0.8% versus 4.2%, *p* = 0.004) in the patients with eyes exhibiting PVR-C at baseline. The greatest retinal break occurred less frequently in the superior retina (71.9% versus 47.2%, *p* < 0.001) and more frequently in the inferior retina (25.2% versus 44.4%, *p* < 0.001) or posterior polar (2.9% versus 8.3%, *p* = 0.009) in patients with eyes exhibiting PVR-C at baseline. The area of detachment was greater (1.9 ± 0.8 quadrants versus 3.3 ± 0.9 quadrants, *p* < 0.001) in the eyes with PVR-C at baseline. Atopic dermatitis (1.1% versus 8.3%, *p* < 0.001), hypotony with an intraocular pressure < 5 mmHg (1.4% versus 17.1%, *p* < 0.001), and choroidal detachment (2.5% versus 25.0%, *p* < 0.001) were also observed more often in eyes exhibiting PVR-C at baseline. The mean AL of the eyes with PVR-N or PVR-B was 25.7 ± 1.9 mm, and that of PVR-C was 24.9 ± 2.1 mm, and the difference between them was statistically significant (*p* = 0.001).

We analyzed the risk of PVR-C using logistic regression analysis adjusted for age and sex in the entire study cohort (Table 2). The following factors were associated with the presentation of PVR-C at baseline: retinal detachment accompanied by a macular hole or dialysis, but not retinal holes or horseshoe tears (OR, 4.70; 95% CI, 2.10 to 10.52; *p* < 0.001), presence of the greatest break in the inferior or posterior polar retina (OR, 2.98; 95% CI, 1.85 to 4.80; *p* < 0.001), greater area of retinal detachment (OR, 5.42; 95% CI, 4.02 to 7.31; *p* < 0.001), atopic dermatitis (OR, 9.40; 95% CI, 3.33 to 26.53; *p* < 0.001), and choroidal detachment (OR, 13.32; 95% CI, 7.20 to 24.65; *p* < 0.001).

We subsequently performed propensity score-matching analyses to investigate the risk factors for the presentation of PVR-C at baseline. The AL was significantly shorter in the eyes exhibiting PVR-C (24.9 ± 2.0, *p* = 0.013) than in those with PVR-N or B (25.9 ± 2.3) at baseline (Table 3). 

Subsequently, we evaluated the risk for PVR-C at baseline in the propensity-matched cohort (Table 4). Logistic regression analysis showed that a shorter AL was a risk factor for presurgical PVR-C at baseline (OR 0.81, 95% CI, 0.69 to 0.96, *p* = 0.015). 

Finally, the influence of baseline findings on the treatment outcome was analyzed in the propensity-matched cohort. Overall, logistic regression analysis showed that the presence of PVR-C at baseline was a risk factor for level 2 or 3 surgical failure (OR, 4.22, 95% CI, 1.12 to 15.93; *p* = 0.034) (Table 5). Moreover, all the eyes with an AL < 25.0 mm that exhibited PVR-C at baseline experienced level 2 or 3 severe surgical failure at 6 months (100%, *p* = 0.016), while the severe surgical failure was not experienced in all the eyes with an AL < 25.0 mm that exhibited PVR-N or PVR-B at baseline (Table 6, Figure 1). This was in contrast to the eyes with an AL ≥ 25.0 mm, where the severe surgical failure was also observed in the eyes that exhibited PVR-N or PVR-B at baseline (42.9%), and the risk did not differ from that of the eyes that exhibited PVR-C (57.1%, *p* = 0.449) (Table 6, Figure 1).

## 4. Discussion

We analyzed the risk factors for the development of preoperative PVR using data from a prospective nationwide multicenter observational study [10,11,12,13] and found that 3.6% of the patients had PVR-C at baseline. A shorter AL was a risk factor for preoperative PVR-C in the propensity-matched cohort. The risk of exhibiting PVR-C at baseline decreased by 19% with a 1-mm increase in the AL. The presence of PVR-C at baseline was a risk factor for level 2 or 3 surgical failure, i.e., the more severe degrees of treatment failure.

Primary analysis using the entire cohort revealed that the characteristics of retinal detachment were risk factors for the development of PVR-C: the area of retinal detachment was a risk factor for preoperative PVR-C with an OR of 5.42. In fact, it was reported that a wide area of retinal detachment was associated with PVR-C [17]. However, it is challenging to discern whether the wide area of retinal detachment was the cause or the result of PVR formation, because PVR formation can expand the detachment. A macular hole or dialysis and the presence of the greatest retinal break in the inferior or posterior polar retina were also risk factors, with ORs of 4.70 and 2.98, respectively. Retinal detachment caused by dialysis and an inferior retinal break may progress slowly, and patients may not become aware of the symptoms at an early stage, which could be associated with the gradual progression of proliferation. A macular hole is reportedly observed frequently in total retinal detachment related to PVR [17]. Previous studies have shown that the baseline presence of PVR-C was related to ocular trauma [6,19] and a past history of uveitis [3,6,19] and inflammation is one of the major components of PVR formation [6,20]. Although only a few patients with PVR-C had experienced trauma to the eyes and none of the patients had a clear history of uveitis, eyes affected by atopic dermatitis, which involves inflammation, were also at risk in the current cohort with an OR of 9.40. Choroidal detachment, which is often reported to be related to preoperative PVR [6,21] was also a risk factor with an OR of 13.32 in the entire cohort; however, choroidal detachment could be the result of chronic inflammation-related hypotony [6,22] which would make it a potential confounding factor.

We performed propensity score matching based on the presence or absence of PVR-C at baseline, taking into account the above-mentioned ORs. We found that a shorter AL was a risk factor for PVR-C, irrespective of surgical invasion; PVR formation may progress easily in eyes with a shorter AL. The impact of a shorter AL on the vitreoretinal condition was discussed in our previous study; the visual outcome of iERM after pars plana vitrectomy was poor in eyes with a shorter AL, although the post-operative best-corrected visual acuity improved in comparison with the preoperative value [15]. We discussed that eyes with a shorter AL may have an undetached or incompletely detached posterior vitreous. This directly mediates the vitreous tractional force to the retina, and the preoperative tractional force by the iERM and/or the intraoperative tractional force exerted on the retina while peeling the iERM may damage the retinal neurons. Vitreous liquefaction is exacerbated in eyes with a long AL [23], and posterior vitreous detachment (PVD) is seen less frequently in eyes with a shorter AL in patients with diabetic retinopathy [24]. In contrast, PVD develops at a significantly younger age in highly myopic eyes compared to the age- and sex-matched non-highly myopic eyes [25]. These facts support the notion that the degree of liquefaction and detachment of the vitreous is lower in eyes with a shorter AL. It is well accepted that a shorter AL is related to choroidal thickening [26,27] which influences the pathogenesis of age-related macular degeneration [26,28,29]. Therefore, AL can affect tissues at both sides of the retina, vitreous and choroid, and various pathogenesis in the eyes.

If PVD is incomplete in eyes with retinal detachment, the RPE cells scattered from the retinal break would easily remain on the vitreoretinal interface to proliferate and develop PVR in eyes with a shorter AL. Moreover, if vitreous adhesion remains, the blood–retinal barrier may be disrupted and the Müller glial cells would be affected, which may easily induce inflammatory cytokines [30,31]. Furthermore, vitreous liquefaction that has not progressed substantially would retard the turnover of inflammatory cytokines, both of which may also promote PVR. In fact, it was previously reported that eyes diagnosed with uveitis had a shorter AL than those without ocular disease, other than cataracts [32].

Generally, eyes with a long AL, which is a risk factor for retinal detachment [10], would be carefully examined when the patients complain of symptoms such as floaters. However, the fundus of eyes with a relatively short AL should also be carefully examined to detect retinal breaks and/or detachment at the early stage and before the development of PVR. The presence of PVR-C at baseline was a risk factor for surgical failure and was particularly associated with severe levels of failure, where the retina is not reattached, at least without tamponade. Although eyes with a longer AL (AL ≥ 25.0 mm) were at a similar risk of surgical failure, irrespective of the presence or absence of PVR-C at baseline, eyes with a shorter AL (AL < 25.0 mm) were at a greater risk of severe surgical failure when they exhibited PVR-C at baseline; while most eyes achieved retinal reattachment at 6 months when they were treated before the onset of PVR-C. Thus, it is vital to detect retinal detachment before the development of PVR-C, particularly in eyes with a shorter AL. 

There were several limitations in the current study. Information on the time-interval between the onset of retinal detachment and baseline-data acquisition was unclear for most patients; although the onset of symptoms was reported for some patients, it was unclear if the symptom reflected the onset of a retinal break, retinal detachment, or macular detachment. Thus, we could not analyze the duration of retinal detachment. If the PVD was incomplete at the biological onset of retinal breaks in the eyes with a shorter AL (as discussed above) the patients may not have complained, or the symptom may have been subtle, thus the clinicians may have disregarded it. Complete fundus examination might not have been performed in such cases, leading to the discovery of the lesions after PVR development. The surgical procedures for treatment were determined by individual surgeons, which might have affected the data on surgical failure; however, all surgeons in the current study were experienced and certified by the Japanese Ophthalmological Society. Additionally, detailed examinations of the posterior vitreous conditions, including partial and/or incomplete PVD, and genetic analyses were not performed, which should be included in future studies.

In conclusion, a shorter AL was a risk factor for the development of preoperative PVR-C and was related to surgical failure. Our results will help to caution clinicians to focus attention in eyes with a shorter AL when retinal breaks and/or detachment are suspected, in order to diagnose retinal detachment before PVR development in routine clinical practice. Further studies that determine whether inflammation is easily induced and/or prolonged in eyes with a shorter AL would be valuable to deepen the understanding of the pathogenesis of PVR, which may help to explore new therapeutic approaches to assist surgical success in the future.

## Figures and Tables

**Figure 1 jcm-10-03944-f001:**
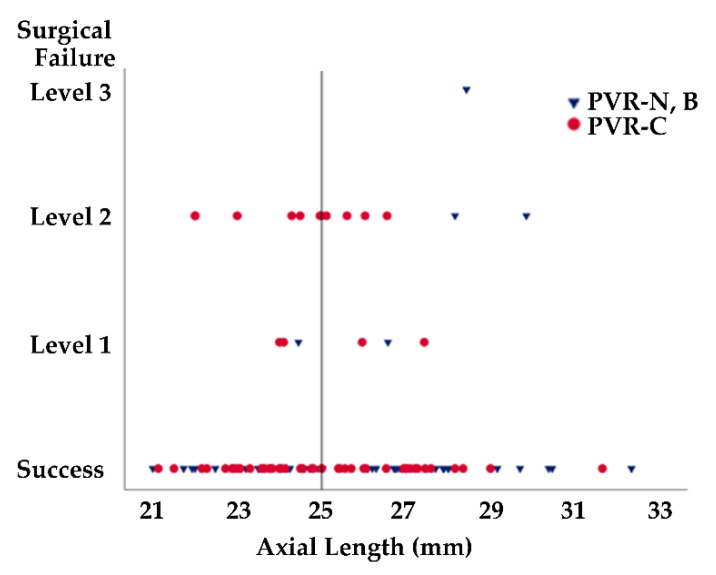
Scatter diagram representing the relationship between axial length and surgical failure/success in the propensity-matched cohort. PVR, proliferative vitreoretiopathy; N, None; B, grade B; C, grade C.

**Table 1 jcm-10-03944-t001:** Baseline characteristics of the participants.

	Total Cohort (*n* = 2013)	PVR Grade		*p*
		PVR-N or B *n* = 1941 (96.4%)	PVR-C *n* = 72 (3.6%)	
Age (years (range))	55.2 ± 15.2 (9–94)	55.2 ± 15.0 (9–94)	55.4 ± 18.8 (10–88)	0.381
Sex (men (%))	1326 (65.9)	1271 (65.5)	55 (76.4)	0.055
Number of retinal breaks (range)	2.0 ± 1.7 (1–26)	2.0 ± 1.7 (1–26)	1.9 ± 1.3 (1–7)	0.646
Type of retinal breaks				
Atrophic holes (eyes (%))	373 (18.5)	355 (18.3)	18 (25.0)	0.150
Horseshoe tears (eyes (%))	1569 (77.9)	1523 (78.5)	46 (63.9)	0.003 **
Macular hole (eyes (%))	52 (2.6)	47 (2.4)	5 (6.9)	0.018 *
Dialysis (eyes (%))	19 (0.9)	16 (0.8)	3 (4.2)	0.004 **
Spread of the greatest retinal break				
≤30 degrees (eyes (%))	1818 (90.3)	1752 (90.3)	66 (91.7)	0.692
>30 degrees (eyes (%))	195 (9.7)	189 (9.7)	6 (8.3)	
Location of the greatest retinal break				
Superior (eyes (%))	1430 (71.0)	1396 (71.9)	34 (47.2)	<0.001 **
Inferior (eyes (%))	521 (25.9)	489 (25.2)	32 (44.4)	<0.001 **
Posterior polar (eyes (%))	62 (3.1)	56 (2.9)	6 (8.3)	0.009 **
Area of detachment (quadrants)	2.0 ± 0.8	1.9 ± 0.8	3.3 ± 0.9	<0.001 **
Secondary retinal detachment				
Atopic dermatitis	28 (1.4)	22 (1.1)	6 (8.3)	<0.001 **
Trauma	21 (1.0)	19 (1.0)	2 (2.8)	0.140
Pseudophakic eyes (eyes (%))	256 (12.7)	245 (12.6)	11 (15.3)	0.507
Hypotony, intraocular pressure <5 mmHg (eyes (%))	39 (1.9)	27 (1.4)	12 (17.1)	<0.001 **
Choroidal detachment (eyes (%))	67 (3.3)	49 (2.5)	18 (25.0)	<0.001 **
Axial length (mm (range))	25.6 ± 1.9 (16.4–33.0)	25.7 ± 1.9 (16.4–33.0)	24.9 ± 2.1 (20.6–31.7)	0.001 **

Data are shown in mean ± SD. Mann-Whitney U test and chi-square test between the groups divided by area of retinal detachment. PVR, proliferative vitreoretinopathy; N, None; B, grade B; C, grade C. ** *p* < 0.01. * *p* < 0.05.

**Table 2 jcm-10-03944-t002:** Risk factors for exhibiting proliferative vitreoretinopathy grade C at baseline.

	Odds Ratio	95% Confidence Interval	*p*
Number of retinal breaks	0.96	0.82 to 1.12	0.572
Type of retinal breaks; Macular hole or Dialysis	4.70	2.10 to 10.52	<0.001 **
Spread of retinal break (s) > 30 degrees	0.78	0.33 to 1.84	0.574
Location of the greatest break; Inferior or Posterior polar	2.98	1.85 to 4.80	<0.001 **
Area of detachment (quadrants)	5.42	4.02 to 7.31	<0.001 **
Atopic dermatitis	9.40	3.33 to 26.53	<0.001 **
Choroidal detachment	13.32	7.20 to 24.65	<0.001 **

Logistic regression analysis adjusted for age and sex. ** *p* < 0.01.

**Table 3 jcm-10-03944-t003:** Characteristics of the patients exhibiting proliferative vitreoretinopathy grade N or B, or proliferative vitreoretinopathy grade C at baseline in the propensity-score matched cohort.

	PVR-N or B (*n* = 64)	PVR-C (*n* = 64)	*p*
Age (years (range))	57.4 ± 16.0	55.9 ± 18.4	0.717
Sex (men (%))	49 (76.6)	48 (75.0)	0.837
Number of retinal breaks (range)	1.7 ± 1.2	1.9 ± 1.3	0.318
Type of retinal breaks			
Atrophic holes (eyes (%))	13 (20.3)	15 (23.4)	0.669
Horseshoe tears (eyes (%))	46 (71.9)	43 (67.2)	0.565
Macular hole (eyes (%))	4 (6.3)	3 (4.7)	0.697
Dialysis (eyes (%))	1 (1.6)	3 (4.7)	0.310
Spread of the greatest retinal break			
≤30 degrees (eyes (%))	59 (92.2)	58 (90.6)	0.752
>30 degrees (eyes (%))	5 (7.8)	6 (9.4)	-
Location of the greatest retinal break			
Superior (eyes (%))	36 (56.3)	34 (53.1)	0.723
Inferior (eyes (%))	23 (35.9)	26 (40.6)	0.585
Posterior polar (eyes (%))	5 (7.8)	4 (6.3)	0.730
Area of detachment (quadrants)	3.2 ± 1.0	3.2 ± 0.9	0.901
Secondary retinal detachment			
Atopic dermatitis	3 (4.7)	5 (7.8)	0.465
Trauma	0 (0.0)	2 (3.1)	0.154
Pseudophakic eyes (%)	17 (26.6)	10 (15.6)	0.129
Hypotony, intraocular pressure <5 mmHg (eyes (%))	6 (9.5)	11 (17.7)	0.180
Choroidal detachment	12 (18.8)	15 (23.4)	0.516
Axial length (mm)	25.9 ± 2.3	24.9 ± 2.0	0.013 *

Data are shown in mean ± SD. Mann-Whitney U test and chi-square test. PVR, proliferative vitreoretinopathy; N, None; B, grade B; C, grade C. * *p* < 0.05.

**Table 4 jcm-10-03944-t004:** Risk for exhibiting proliferative presurgical grade C at baseline in the propensity-score matched population.

	Odds Ratio	95% Confidence Interval	*p*
Axial length	0.81	0.69 to 0.96	0.015 *

Logistic regression analysis in the propensity-score matched population. * *p* < 0.05.

**Table 5 jcm-10-03944-t005:** Risk for surgical failures (levels 2 or 3) in the propensity-score matched population.

	Odds Ratio	95% Confidence Interval	*p*
PVR-C	4.22	1.12 to 15.93	0.034 *

Logistic regression analysis in the propensity-score matched population. PVR-C, proliferative vitreoretinopathy grade C. * *p* < 0.05.

**Table 6 jcm-10-03944-t006:** Risk for surgical failures (level 2 or 3) according to axial length in the propensity-score matched population.

	PVR-N or B	PVR-C	*p*
Axial length < 25.0 mm	0 (0.0)	7 (100.0)	0.016 *
Axial length ≥ 25.0 mm	3 (42.9)	4 (57.1)	0.449

Fisher’s exact test. PVR, proliferative vitreoretinopathy; N, None; B, grade B; C, grade C. * *p* < 0.05.

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
