# Peer review of "Shorter Axial Length Is a Risk Factor for Proliferative Vitreoretinopathy Grade C in Eyes Unmodified by Surgical Invasion"

_jcm, 2021, doi:10.3390/jcm10173944_

Round 1
Reviewer 1 Report
I would like to congratulate the authors for this manuscript, as it gives useful information for the retinal specialist.
There are only sone minor revisions requires:
The text needs proper editing as the tables and images are many times superimposed and it damages its readability.
In the Methods section, please state which PVR classification is used.
Also in the Methods section, point 2.4. “Definition of primary success and failure” needs some revision. In this paragraph the authors briefly describe the definition of success rate per the European vitreo-retinal society retinal detachment study. However, the levels are inverted, eg in the EVRS study level 1 signifies inoperable eyes with detached retinas and in this study it is level 3. Is this an error, or was it done on purpose and why. Also, as this is a general medicine paper, it is important to state that the level 1 (or level 3 as per EVRS) are eyes in which silicone oil has been removed.
In the Results section, Scatter diagram is difficult to interpret and doesn’t offer any useful additional information. Consider if is worth maintaining.
In the Discussion section, the last sentence of the fourth paragraph (lines 234-237) is not very clear on its conclusion. Do the authors want to imply that choroidal thickening might be related to PVR in the short AL eyes with RRD? What is their hypothesis to explain this?
Author Response
I would like to congratulate the authors for this manuscript, as it gives useful information for the retinal specialist.
Thank you for your understanding our study and careful review.
There are only sone minor revisions requires:
The text needs proper editing as the tables and images are many times superimposed and it damages its readability.
Thank you for your valuable advice. We edited the tables and images to improve readability. It is requested that tables and images are superimposed in the text according to the author guideline of this journal.
In the Methods section, please state which PVR classification is used.
Thank you for your advice. We have used PVR classification reported by Machemer et al in 1991, and added a reference in the revised text as follows;
Line 126
We performed logistic regression analysis to prepare the propensity score-matched cohort according to the presence or absence of retinal detachment exhibiting PVR grade C (PVR-C) diagnosed using updated Retina Society Classification published in 1991.[1]
Reference 1
Machemer, R.; Aaberg, T.M.; Freeman, H.M.; Irvine, A.R.; Lean, J.S.; Michels, R.M. An updated classification of retinal detachment with proliferative vitreoretinopathy. Am J Ophthalmol 1991, 112, 159-165, doi:10.1016/s0002-9394(14)76695-4.
Also in the Methods section, point 2.4. “Definition of primary success and failure” needs some revision. In this paragraph the authors briefly describe the definition of success rate per the European vitreo-retinal society retinal detachment study. However, the levels are inverted, eg in the EVRS study level 1 signifies inoperable eyes with detached retinas and in this study it is level 3. Is this an error, or was it done on purpose and why. Also, as this is a general medicine paper, it is important to state that the level 1 (or level 3 as per EVRS) are eyes in which silicone oil has been removed.
We understand your concern. However, the study had decided to define the criteria as this, from the beginning of this prospective study, and also used our previous paper by Baba et al. Also, Silicone oil removal was planned within 6 months if no other additional procedures were required, and the surgery to remove silicone oil was not counted as an additional surgery. We inserted a phrase to avoid confusion, added explanation regarding silicone oil removal, and the reference of Baba’s paper as follows;
Line 116
Briefly, level 1 signifies the requirement of additional surgeries for recurrent retinal detachment after the initial surgery and that the retina is attached at month 6; level 2 signifies the requirement of sustained silicon oil tamponade at month 6, and level 3 signifies the persistence of retinal detachment with or without additional surgeries at month 6; the most severe failure was level 3.[11] Silicone oil removal was planned within 6 months if no other additional procedures were required, and the surgery to remove silicone oil was not counted as an additional surgery.[11]
In the Results section, Scatter diagram is difficult to interpret and doesn’t offer any useful additional information. Consider if is worth maintaining.
Thank you for your advice. We considered that not only the average but the distribution of the failure cases according to the AL would be helpful to understand the actual situation, and included the data in the revised text.
In the Discussion section, the last sentence of the fourth paragraph (lines 234-237) is not very clear on its conclusion. Do the authors want to imply that choroidal thickening might be related to PVR in the short AL eyes with RRD? What is their hypothesis to explain this?
No. We wanted to describe that AL has been thought to relate to the choroidal and retinal lesions, moreover, the current study, together with previous studies referred in the text, proposed that it may be also related to the vitreous condition. We revised the text as follows;
Line 251
It is well accepted that a shorter AL is related to choroidal thickening,[26,27] which influences the pathogenesis of age-related macular degeneration.[26,28,29] Therefore, AL can affect tissues at both sides of the retina, vitreous and choroid, and various pathogenesis in the eyes.

Reviewer 2 Report
The authors have associated several risk factors for the development of PVR grade C, such as an axial length (AL) of < 25mm, retinal detachment accompanied by a macular hole or dialysis, but not retinal holes or horseshoe tears, presence of the greatest break in the inferior or posterior polar retina, greater area of retinal detachment, choroidal detachment and atopic dermatitis. In addition, they evaluated the risk of PVR-C at baseline in the propensity-matched cohort and found that AL was a risk factor for pre-surgical PVR-C at baseline.
In addition, surgical failure increased with baseline PVR-C and eyes with AL < 25mm, whereas in eyes with AL ≥ 25mm surgical failure of eyes with PVR-N or PVR-B did not differ to the risk of eyes with PVR-C.
The topic and the findings of the manuscript are novel and are of clinical interest. The manuscript is structured and understandable with a big patient population and good statistical work up.
Minor points:
Please describe in the introduction the different PVR grades and classifications.
Author Response
The authors have associated several risk factors for the development of PVR grade C, such as an axial length (AL) of < 25mm, retinal detachment accompanied by a macular hole or dialysis, but not retinal holes or horseshoe tears, presence of the greatest break in the inferior or posterior polar retina, greater area of retinal detachment, choroidal detachment and atopic dermatitis. In addition, they evaluated the risk of PVR-C at baseline in the propensity-matched cohort and found that AL was a risk factor for pre-surgical PVR-C at baseline.
In addition, surgical failure increased with baseline PVR-C and eyes with AL < 25mm, whereas in eyes with AL ≥ 25mm surgical failure of eyes with PVR-N or PVR-B did not differ to the risk of eyes with PVR-C.
The topic and the findings of the manuscript are novel and are of clinical interest. The manuscript is structured and understandable with a big patient population and good statistical work up.
Thank you for your understanding our study and careful review.
Minor points:
Please describe in the introduction the different PVR grades and classifications.
Thank you for your advice. We included the topic in the introduction section as follows;
Line 54
PVR is classified by clinical signs and can progress from grade A to C; grade A is defined by vitreous haze and pigment clumps, grade B is by wrinkling of inner retinal surface, retinal stiffness, vessel tortuosity, rolled and irregular edge of retinal break, and decreased mobility of vitreous, and grade C is by full-thickness retinal folds and/or subretinal strands posterior to and/or anterior to equator.[1,7]
